# The Role of Novel Bladder Cancer Diagnostic and Surveillance Biomarkers—What Should a Urologist Really Know?

**DOI:** 10.3390/ijerph19159648

**Published:** 2022-08-05

**Authors:** Rafaela Malinaric, Guglielmo Mantica, Lorenzo Lo Monaco, Federico Mariano, Rosario Leonardi, Alchiede Simonato, André Van der Merwe, Carlo Terrone

**Affiliations:** 1Department of Urology, IRCCS Policlinic Hospital San Martino, 16132 Genoa, Italy; 2Dipartimento di Scienze Chirurgiche e Diagnostiche Integrate (DISC), University of Genoa, 16132 Genoa, Italy; 3Department of Urology, Casa di Cura Musumeci GECAS, 95030 Gravina di Catania, Italy; 4Department of Surgical, Oncological and Stomatological Sciences, University of Palermo, 90133 Palermo, Italy; 5Department of Urology, Tygerberg Academic Hospital, Stellenbosch University, Cape Town 7600, South Africa

**Keywords:** bladder cancer, biomarkers, diagnosis, surveillance

## Abstract

The aim of this review is to analyze and describe the current landscape of bladder cancer diagnostic and surveillance biomarkers. We researched the literature from 2016 to November 2021 to find the most promising new molecules and divided them into seven different subgroups based on their function and location in the cell. Although cystoscopy and cytology are still the gold standard for diagnosis and surveillance when it comes to bladder cancer (BCa), their cost is quite a burden for national health systems worldwide. Currently, the research is focused on finding a biomarker that has high negative predictive value (NPV) and can exclude with a certainty the presence of the tumor, considering missing it could be disastrous for the patient. Every subgroup has its own advantages and disadvantages; for example, protein biomarkers cost less than genomic ones, but on the other hand, they seem to be less precise. We tried to simplify this complicated topic as much as possible in order to make it comprehensible to doctors and urologists that are not as familiar with it, as well as encourage them to actively participate in ongoing research.

## 1. Introduction

The term ‘biomarker’ has been used in scientific language since the 1970s and the most accepted definition of it is ‘a defined characteristic that is measured as an indicator of normal biological processes, pathogenic processes or responses to an exposure or intervention’. Biomarkers can be found at molecular, cellular and tissue levels [1].

The advantage of biomarkers is their simplicity in the method of clinical endpoint measurement, and their reproducibility, along with repeated, immediate analysis and cost-effectiveness when compared to other diagnostic and staging tools [1,2].

When both sexes are considered, bladder cancer (BCa) is currently one of the most common cancers worldwide, sitting at seventh place. Around 75% of patients present with a disease confined to the mucosa or submucosa [3], but they often recur [4,5]. At this moment, urinary cytology and cystoscopy represent the standard of care in both diagnosis and surveillance [6,7]. Cytology, although having a great sensitivity for high-grade tumors (84%), is not performing as well for low-grade tumors (16%) [8]. Furthermore, the interpretation of it is user-dependent and can be easily tempered [9,10]. For example, reactive urothelial atypia during cystitis is difficult to recognize when compared with recurrent CIS [11]. Moreover, there is a high proportion of cytology samples that are categorized as “atypical” [12]. Cystoscopy, in contrast, is sensitive for papillary lesions but not as much for flat ones. It also bares certain risks, especially in elderly and fragile patients [6]. Finally, in addition to the guidelines, surveys have shown that patients prefer undergoing cystoscopy, accepting its risks, than having uncertainties regarding their oncological status by taking noninvasive blood or urine tests [13,14]. Actually, some patients undergo cystoscopies more frequently than recommended, associating this kind of surveillance with increased medical costs [15,16,17,18].

What we consider an effective assessment that could greatly benefit current practice, is a low-cost, rapid and effective test with high sensitivity and negative predictive value. By ruling out the presence of bladder cancer, it would help patients avoid the cystoscopy [19,20,21,22,23].

Several urinary tests have been developed during recent years, but none of the novel biomarkers met the criteria to be proposed as a routine practice in diagnosis and follow-up. The main reasons are lower specificity when compared to cytology, clinical context dependency when applicated, and complicated laboratory methods, although there are a few that have yielded a high negative predictive value (NPV) [3].

The aim of this review is to summarize current evidence and provide to urologists an overview of this sometimes quite complicated landscape.

## 2. Materials and Methods

### 2.1. Literature Research Strategy

We conducted systemic research on Pubmed/Medline and Web of Science using the following Medical Subject Headings (MeSH) terms: ((“Biomarkers, Tumor” [Mesh]) AND (“Diagnosis” [Mesh]) OR (“Early Detection of Cancer” [Mesh]) AND (“Urinary Bladder Neoplasms” [Mesh])) from 2016 to November 2021, wanting to evaluate the newest biomarker studies and proposals.

The updated Preferred Reporting Items for Systematic reviews and Meta-Analyses (PRISMA) guidelines were followed for this systematic review [24]. The search was performed independently by two researchers (R.M. and G.M.), and any disagreement resolved by a third independent researcher (L.L.M.). The initial screening was done based on titles and abstracts.

### 2.2. Inclusion and Exclusion Criteria

All papers published after 2016 concerning studies conducted on diagnostic and surveillance biomarkers for bladder cancer were considered for the review.

We included full text articles written in the English language, with a cohort of at least 10 patients, aged 18 and older. After reviewing each manuscript, we also searched its references for any relevant manuscripts missed. We excluded the systematic/narrative reviews, meta-analyses, nonrelevant articles, editorials or letters to the editor, case report articles, and studies on the accuracy of biomarkers in assessing prognosis, monitoring treatment response or therapy guided.

### 2.3. Data Extraction Design

Biomarkers were classified into 7 different categories: altered proteins, genomic mutations, circulating free nucleic acids, epigenetic changes and transcriptomic alterations, metabolic-induced changes, combination of more than one category and improving conventional cytology.

Two investigators (R.M. and G.M.) extracted details about study design, biomarkers studied, evaluated tests, population studied and its characteristics, bladder cancer grade and stage and results. A third investigator (L.L.M.) verified extractions for accuracy and was consulted for discrepancies, which were resolved through consensus.

We constructed Table 1 comparing sensitivities and specificities of the biomarkers from the published articles (data extracted from the results section of each manuscript).

### 2.4. Statistical Analysis

Data were entered into a Microsoft Excel (Version 2019) database. Descriptive statistics were calculated for all demographic, clinical and follow-up variables and reported as median or as a proportion with percentage.

## 3. Results

By searching the above-mentioned MeSH terms, we identified 778 articles in Pubmed/MEDLINE and 49 articles in the Web of Science database; 10 articles were present in both databases, therefore, the total number of abstracts reviewed was 817.

Subsequently, 691 articles were excluded after full text review because they did not meet inclusion criteria, and 36 articles were added from additional sources (references of the reviewed manuscripts). Therefore, 118 articles were eligible for the final analysis.

The PRISMA flowchart is shown in Figure 1.

This review was registered in The International Prospective Register of Systematic Reviews (PROSPERO) database (the protocol No. CRD42021270777).

### 3.1. Altered Proteins

Some authors believe that the detection of altered proteins is the most appropriate due to minimal sample processing and lower costs, without requiring molecular target amplification. Usually, altered proteins are detected in assays and by spectrometry [25,26].

For quite some time, scientists were oriented towards investigating filaments of cytoskeleton, the **cytokeratins**. Their expression is dependent on the cellular differentiation and its degree, so they represent a good biomarker for distinguishing carcinomas [27]. Four types of cytokeratins were tested as potential urinary biomarkers: 8, 17, 18 and 20. The only commercialized test using cytokeratins (8 and 18) is UBC rapid test. It provides diagnosis after 10 min using a photometric reader. Although very fast, practical and cheap, the UBC test’s sensitivity (30–87%) and specificity (63–91%) [28,29,30,31,32] are quite limited. It was, however, proved to be a useful in adjunct to the cytology with immunostaining [33,34].

Another structural protein proposed as a biomarker is **nuclear matrix protein 22**. During the cell-division cycle, NMP22 is released from the apoptotic cells and its presence increases significantly in BCa patients [35]. NMP22 was approved as a diagnostic biomarker by the Food And Drug Administration (FDA) (Bladdercheck), but this biomarker is not recommended for routine use because of its variable sensitivity (37.9–88.5%), depending mostly on the grade and stage, and specificity (65.2–96.9%) in bladder cancer detection [36].

Onsidering cellular signaling proteins such as **Orsomucoid-1** (acute-phase protein [37]), **Osteonectin, BM40 or Secreted Protein Acidic and Rich in Cystein** (SPARC—Extracellular matrix protein that promotes cellular adhesion [38]), **Apurinic/apyrimidinic endonuclease 1/redox factor-1** (DNA repair and activation of transcriptional factors [39]), **soluble Fas** (programmed cellular death via Fas/APO-1/CD95 [40]), **Aurora kinase A** (AURKA—Assembly of mitotic spindle and facilitation of the mitosis [41]), **Serum Irisin^,^** (activator of gene promoters for the browning of the white adipose tissue [42]), **Amplified in breast cancer 1** (AIB1—Transcriptional activity [43]) in combination with **EIF5A2** (protein synthesis and in translation elongation [44]), when employed as a diagnostic biomarkers yielded variable sensitivities and specificities, ranging from 39% to 93% and 70% to 94%, respectively [45,46]. Although outperforming urine cytology when it comes to sensitivity, some were largely influenced by the presence of hematuria, or did not achieve satisfactory NPV to be considered for the everyday practice use.

**LIM and SH3 protein 1 (LASP1)**, adhesion proteins, were also tested, but subsequently excluded as diagnostic biomarkers because of false positives high rates [47].

Protein biomarkers are much more likely to be diagnostic if used in pairs or in assays. In fact, **tissue-like isoforms** (unphosphorylated and Ser253- and Ser258-phosphorylated, involved in blood clotting via extrinsic pathway and angiogenesis) were shown to be quite specific for the BCa cohort. With an optimum cut-off established, this assay yielded high specificity and NPV, but it needs further investigatory trials [48]. On the other hand, a multiplex immunoassay containing **10 protein biomarkers** (angiogenin, apolipoprotein E, alpha-1 antitrypsin, carbonic anhydrase 9, interleukin 8, matrix metallopeptidase 9 and 10, syndecan 1, plasminogen activator inhibitor 1 and vascular endothelial growth factor A), commercially launched as Oncuria, achieved an overall sensitivity of 87% and specificity of 92%, but when corrected for three demographic factors (sex, age, race) the overall sensitivity and specificity reached 93%, with positive predictive value (PPV) of 65% and NPV of 99% [49,50,51].

### 3.2. Genomic Mutations

Cancer is a result of somatic mutations in which clonal lineage gains mutations that allow cell survival in a hostile environment [52]. By using The Cancer Genome Atlas (TCGA) we can now identify and correlate different genes that are most frequently mutated to the specific types of cancer [53].

One of the most important characteristics of the cancerous cells is to proliferate indefinitely. Deoxyribonucleic acid (DNA) polymerase limits normal cell proliferation by shortening chromosomes, but if telomeres are added, the cell continues to duplicate. Telomeres can be shortened by telomerase, an enzyme that is regulated by TERT (Telomerase reverse transcriptase) promoter. Cancer cells do not downregulate telomerase [54].

Meanwhile, Fibroblasts Growth Factor Receptors (FGFR) are transmembrane receptors kinases that, when activated, promote cell proliferation, differentiation, growth, migration and apoptosis [55]. FGFR3 mutation is usually associated with shorter time to recurrence in low-grade disease [56].

One of the novel urine-based assays that detects mutations in three genes (TERT, FGFR3 and KRAS), FDA-approved Uromonitor, was investigated in two major studies. With sensitivity that varied from 79% to 98%, specificity 62–90% and NPV 86–99%, the test showed potential as an alternative to the current follow-up methods, to be confirmed in larger cohorts of BCa patients [57,58].

Multicentric and prospective study done by Beukers et al. evaluated **the combination of TERT, FGFR3 and OTX1 mutations panel**. During surveillance 57% of recurrences were identified, with specificity value of 59%. The majority of tumors missed were low-grade tumors (45–49%), whilst T1, T2-4, high-grade G3 tumors were identified in 88%, 77% and 83% of patients during follow-up. FGFR3 mutation can be combined with other common mutations, such as Cyclin D3, or in an assay containing **TP53, PIK3CA, ARID1A, STAG2 and KTM2D**, yielding sensitivity, specificity and NPV of 73–95%, 85–90% and 0.82, respectively [59,60,61,62].

**TERT and PLEKHS1 promoter mutations** were demonstrated to be more sensitive than cytology alone, especially in low-grade tumors. Overall sensitivity, specificity and NPV varied, reaching values of 91–96.7%, 96–100% and 87.5% [62].

**ASPM**, or the abnormal spindle-like microcephaly associated gene, involved in mitotic spindle formation, overexpresses in neoplastic tissues, while **TEF**, or thyrotrophic embryonic factor, is involved in transcription by binding DNA, hence controlling cell cycle [63]. Saleh et al. analyzed the expression of these two genes in BCa and found ASPM upregulation and TEF downregulation in cancerous tissue when compared to the healthy one. ASPM gene showed sensitivity of 95.56% and specificity of 86.67%, whilst TEF 91.11% and 71.11%. Furthermore, these two genes have the ability to predict occurrence of metastatic lesions, being significantly over or under expressed in high-grade tumors, and smokers [64].

Further, minichromosome maintenance protein 5 (**MCM5**) is an essential protein when it comes to the initiation of DNA replication process. It is usually present only on the surface of actively dividing cells [65]. ADXBLADDER MCM5 is an ELISA test that utilizes detection of MCM5 via MCM5 antibodies in urine [65]. This test was evaluated in a multicentric, blinded, prospective study where it outperformed cytology’s sensitivity. It yielded sensitivity of 63–79%, specificity of 65–88% and a negative predictive value of 87–99%. In addition, **ADXBLADDER** does not require an interpretation of an expert pathologist and the result is available in 3h. However, in some studies, pTa low-grade cases were regarded as negative for BCa, increasing, therefore, NPV. Nevertheless, according to these results, a negative test could indicate the absence of the recurrence [65,66,67,68,69,70].

Lastly, when repeated alterations in the genome occur, they can cause **microsatellite instability**. Consequently, frameshift mutations occur and stimulate tumor growth [71]. Loss of heterozygosity (LOH) is another molecular change that affects gene loci, and it occurs when one of the microsatellite alleles is present in the normal DNA but not in the paired tumor DNA. When profiling for microinstability an Egyptian study found a concordance between urine and tumor tissue samples regarding D16S476, D9S171, FGA, and ACTBP2 alterations (MSI and/or LOH), therefore, they evaluated it as a potential biomarker [72].

### 3.3. Circulating Free Nucleic Acids

**Nucleic acids** are present in bodily liquids independently of cells. Cell-free nucleic acids (CfNAs) can contain nuclear DNA, mitochondrial DNA, and all types of RNA. CfDNA can be found free, vesicle-bound or in DNA-macromolecular complexes, whilst RNAs in extracellular vesical membranes (EVM) or form ribonucleoprotein complexes. In addition, circulating DNA-protein complexes are double-strand DNA (dsDNA) fragments wrapped around the histone protein complex, virtosomes and are released during apoptosis. Mitochondrial DNA, in contrast, is secreted exclusively in EVMs. Various types of RNAs are divided in two major groups: coding (for proteins) and noncoding [73].

**Tumor cell-free DNA** can be found in urine of patients, so its quantitative analysis has been explored as a possible biomarker for BCa detection. Brisuda et al. found that urine-derived cfDNA was not significantly more advantageous than urine cytology, having sensitivity and specificity of 20.7 and 91.2%, respectively, in low-grade and 59.5 and 91.2%, respectively, in high-grade tumors [74]. On the other hand, serum and urine exosomes performed quite well, showing specificity of 100% in serum and 83.3% in urine, whilst sensitivity of 92.6% in urine and 82.4% in serum. The correlation between exosomes levels and invasiveness of bladder cancer was also noted [75].

CxBladder is an FDA-approved, multiplex and multi-assay urinary test that quantifies **messenger RNA (mRNA)**. This test can detect cancer in the patients’ urine and estimate the probability of its presence by extracting and quantifying these five mRNAs: MDK (migration and angiogenesis of the cancerous cells), HOX13 (affects cell differentiation), CDC2 (completion of cell cycle), IGFBP5 (inhibits the apoptosis) and CXCR2 (crucial for inflammatory response). Based on the clinicians’ requirements Cxbladder offers three types of urinary tests: Triage (high NPV to rule out the presence of cancer in patients with hematuria), Detect (in adjunct to cystoscopy and other diagnostic procedures aiming to detect recurrent bladder cancer with higher accuracy) and Monitor (presents high sensitivity and it should be able to replace more invasive surveillance procedures) [75]. With a sensitivity of 91%, specificity of 61% and NPV of 98% for detecting high-grade recurrence, it was found that CxbM could be useful as a rule-out test to identify the patients who are low-risk and do not require cystoscopy. In total, 77.8% of patients were managed in a safe and feasible manner like that, repeating cystoscopy every two years. Moreover, 35% of unnecessary cystoscopies were spared [76,77,78,79].

Alternatively, Xpert Bladder is a novel, extremely practical and low-cost urinary test for BCa detection. By quantifying five target **mRNAs** (CRH, IGF2, UPK1B, ANXA10 and ABL1), it automatically prepares the sample, amplifies and targets the desired sequences to give a positive or a negative response. Overall sensitivity of the test was 63–80%, specificity 73–81% and NPV 92%. On the other hand, when stratified by grade, it achieved NPV of 99% for high-grade recurrences, so this could imply less cystoscopies in patients harboring this kind of tumors [66,80,81,82,83,84,85,86,87,88,89].

Other researched mRNAs present in higher levels of BCa patients’ urine are Urine Ubiquitin Conjugating Enzyme E2 C (UBE2C—A member of the anaphase promoting cyclosome), QGAP3 (cell proliferation, cytoskeleton remodeling, cellular adhesion and signaling), and N-Myc downstream-regulated gene 2 (cell proliferation and apoptosis promotion). UBE2C mRNA was found to be particularly useful in discriminating the origin of hematuria [90], QGAP3 in differential diagnosis in gross and microhematuria, whilst N-Myc mRNA levels were highly correlated with grade and stage of the disease. Overall, singularly, these mRNAs achieved sensitivity 80–96.2%, and specificity that varied from 60.2 to 90.7%) [90,91,92].

### 3.4. Epigenetic Changes and Transcriptomic Alterations

**Micro RNAs (miRNAs) are small non-coding RNAs** transcribed from DNA sequences. They usually interact with untranslated regions (regions that do not carry information from DNA to the ribosomes, the site of the protein synthesis). However, recent research suggests that miRNAs might have hormone-like activities by influencing protein levels of the target mRNAs without affecting the gene sequences [93].

Usually, epigenetic biomarkers are not used singularly, but in assay, as multiple miRNAs combined in a panel showed greater diagnostic performance [94]. Study done by Urquidi et al. determined 25 target miRNAs by using multivariate model and concluded that the diagnostic ability of each biomarker increases by shrinking the regression coefficients of the parameters [95]. They can be targeted by using next generation sequencing (NGS), although, with this technology some discrepancies were observed when compared with real-time polymerase chain reaction (PCR). Considering these findings, it cannot be stated with certainty which method is best for detecting urinary miRNAs [96].

A lot of miRNAs subtypes have been studied recent years, singularly and in assays, and their altered concentrations in the patients’ urine, with quite high sensitivity and specificity [94,97,98,99,100,101,102]. Interestingly, we can combine them with other diagnostic tools, such as suprapubic ultrasound. In fact, when miRNA-192 was combined with 2D bladder ultrasound the sensitivity and specificity were 93.2% and 96.7%, respectively, not significantly different from that of cystoscopy [100]. Moreover, in a study that tested a diagnostic assay containing 12 miRNAs, it was hypothesized that it could lower the cystoscopy rates by 30% [103], while a panel containing only 7 miRNAs significantly outperformed cytology in low-grade tumors (sensitivity = 80.88% and specificity = 91.67%) [98]. Juracek et al. concluded that only one pair of miRNAs allows discrimination between grades and stages and was significantly decreased in patients in postoperative period if disease-free [99]. MiRNAs can be retrieved from urine, but also from the patients’ serum. The combination of seven serum miRNAs resulted to be quite characteristic of BCa, yielding overall high sensitivity (95–98%) and specificity (87–91%), and, with two specific miRNAs, miRNA-6087 and miRNA6831-5p, showed remarkable diagnostic power for detecting BCa and distinguishing it from other solid tumors [104]. The three miRNAs serum panel also showed quite a promise, although not as such as a seven-component panel [102].

On the other hand, studies have confirmed the importance of **DNA methylation regulation** in cancer transformation, preservation, and progression. If DNA methylation becomes dysregulated, it influences not only the integrity of the genome, and thus dysregulation of gene expression, but also the immune modulation-response [105].

Urine-based DNA methylation test containing a panel of **15 (unpublished) genes**, commercially known as **Bladder EpiCheck^®^**, was the first test used for monitoring bladder cancer patients. Based on the methylation levels present in the urine, the test gives a numeric value between 0 to 100 (EpiScore), considering it positive for BCa if Episcore results over 60. The test yielded an overall sensitivity of 57–85% (91.7% when low-grade tumors were excluded) and a specificity of 80–88.0%. After excluding low-grade recurrent tumors, sensitivity and NPV increased, with NPV up to 99% [106,107,108,109,110,111]. One study compared Bladder EpiCheck and Bladder Xpert Monitor performance (measures mRNA in the patients’ urine—ABL1, CRH, IGF2, UPK1B and ANXA10) with those of cytology and found that these new tests outperform cytology when it comes to sensitivity, but do not reach its excellence when it comes to specificity. NPV was very similar for the three tests, but when combined, two novel tests detected 79.35% of the tumors: 70.37% low grade and 92.11% high grade [66].

Combining genes such as TWIST1, OTX1 (Homebox 1 codes for a transcriptional protein), and TERT in combination with FGFR3 [60,112,113,114,115,116] yielded the most success when evaluating abnormal methylation of tumor suppressor genes [117]. By testing them, a high-throughput microdroplet-based PCR amplification system (UroMark) [118,119] showed sensitivity of 87–98%, specificity 96% and NPV of 97% [119].

RNA has an important role of being intermediary between DNA and encoded proteins, and although only 1.5% of human DNA encodes for protein genes, 90% of the genome is actively transcribed. This process is extremely complex and consists of **various types of RNA**. For example, long non-coding RNAs (lncRNAs) are involved in majority of cellular cycle processes and regulate diverse functions such as controlling apoptosis, cell death and cell growth [120].

Firstly, Sin et al. applied RNA sequencing tool for discovering bladder-cancer specific, 3-component mRNA panel, ergo urinary concentration of ROBO1 (promoter of tumor angiogenesis, BCa specific), WNT5A (regulation of cell polarity and migration, BCa specific) and CDC42BPB (the reference gene) mutational load. This assay was evaluated in an independent cohort and later compared to conventional cytology. It yielded overall sensitivity and specificity of 83% and 89%, respectively, and was more sensitive than CxBladder in low-grade tumors [121].

Alternatively, lncRNAs can be identified in the patients’ urine and serum, and they too preferably in panels. LncRNAs serum panels, developed and tested, achieved AUC of 0.857 and 0.826, significantly higher than that of urine cytology. Additionally, the researchers noticed that non-muscle-invasive BCa patients with high UBC1 lncRNA expression had significantly lower recurrence-free survival. Ultimately, overall sensitivity was 80%, whilst specificity ranged from 73% to 75% [122,123].

Other panels, detecting urinary lncRNAs, such as UCA1-201, UCA1-203, MALAT1 or HOTAIR lncRNAs, yielded greater sensitivity and specificity (43.5–100% and 72–100%, respectively) than those targeting serum lncRNAs, but those too need further research and trials before taken in consideration for any further clinical application [122,123,124,125,126,127,128,129].

Transcriptomic alterations are quite powerful diagnostic tool as well, so much that the expression of only two genes, IGF2 and MAGEA3, yielded almost the same sensitivity and specificity as the combination with others, 81% and 91%, respectively. These two genes make up the core of BCa, and were validated in a prospective, blinded study [130].

### 3.5. Metabolomics and Metabonomics

Metabolites are substrates and products of metabolism that are involved in cellular cycle and influence cell functions [131].

Metabolomics is the profiling of metabolites in biological fluids, cells, and tissue, routinely applied as a biomarker [132].

Metabonomics, on the other hand, is the measurement of changes across the metabolome, with respect to time, due to an intervention [132].

Numerous metabolites have been studied for BCa detection and monitoring, and different methods employed to evaluate their presence and/or alterations in biologic liquids: nuclear magnetic resonance, liquid chromatography, mass spectrometry, electrophoresis and fluorescence are only some of them [133,134,135,136,137,138,139,140]. Raman chemometric urinalysis was proposed as a simple, low-cost and rapid screening test evaluating characteristic BCa metabolic signature, as well as Fourier infrared spectroscopy on the bladder wash, yielding sensitivity of 82.4% and specificity of 79.5% [134,135]. Other substances considered as diagnostic biomarkers are dimethyl amine and valine metabolites, some fatty acids, histidine and retinol metabolites, purine metabolism end-products, peptide and collagen fragments, as well as few essential amino acids that varied greatly in pre- and post- operative period, and their levels were found to be correlated with tumor grade. Overall sensitivities varied from 51–98%, specificities from 51–77.3% and NPV 91.2–93.6% [133,134,135,136,137,138,139,140]. Interestingly, a panel of six ions was also considered, of which the most representative was imidazoleacetic acid, but resulting in lower sensitivity and specificity than other metabolites, and thus, not further investigated [136].

Metabolites can be measured also in patients’ serum, in fact five peptides that were evaluated as a potential BCa serum diagnostic biomarker achieved sensitivity of 93.75% and specificity of 96.77%, but they also need ulterior clinical validation [140].

### 3.6. Combination of Different Categories

It was hypothesized that the combination of urine exosomal miRNA expressions and alterations in certain genes, as a panel for BCa detection and diagnosis would result in more accuracy when compared to only one category. Amuran et al. proposed measuring concentrations of urinary exosomal miR-19b1-5p, 21-5p, 136-3p, 139-5p, 210-3p expressions and urinary BLCA-4, NMP22, APE1/Ref1, CRK, VIM and creatinine concentrations as a diagnostic panel. They found that it was able to diagnose BCa with 80% sensitivity and 88% specificity as well as differentiate low risk patients from healthy controls [46]. On the other hand, combining only miRNA663 and methylated VIM diagnostic values were slightly inferior, 92.6% sensitivity and 90% NPV [141].

Furthermore, conventional cytology can be bettered by assessing, at the same time, methylation of some genes, such as NID2 and TWIST1 combined, or using the panel of SALL3, CFTR and TWIST1 [142,143] the same as pairing altered methylation with TERT and FGFR3 mutations [144].

The combination of 3 miRNAs and 2 lncRNAs, overly altered in BCa, was also evaluated, but did not display optimal diagnostic features (AUC = 0.910; sensitivity = 87.2% and specificity = 83.3%) [145], the same as coupling metabolomics assay and 5 target mRNAs (Xpert panel) resulting in NPV of 98.2%, not much better than when used singularly [146].

Finally, Fouad et al. tried to assess diagnostic power of protein and gene expression levels of tissue inhibitor of metalloproteinase-2 (TIMP-2), matrix metalloproteinase-2 (MMP-2), and MMP-9 in urine and blood. The researchers reported excellent specificities, sensitivities and NPV, but the panel has not been ulteriorly tested [147].

### 3.7. Improving Conventional Cytology

It was suggested that by adding the immunohistochemical staining to the cytology its diagnostic value could improve. Courtade-Saidi et al. added coloration for p53 and ki-67 when analyzing cells from bladder, upper urinary tract and neobladder. When comparing specificity, the results were 76.7% for cytology alone, 93.2% for cytology with p53, 88.0% for cytology with Ki-67, and 97.5% for cytology with p53 and Ki-67, without penalizing the sensitivity for all grades. Combined cytology and p53 was better for detecting low-grade tumors, whilst cytology and ki-67 was optimal for detecting high-grade tumors [148]. Other biomarkers proposed as an adjunct to conventional cytology such as CD20 and AMARC, CD20 and p53 or p53 combined with MCM5, MCM3 and ki67, or even dual-labeling ki67/p16, when correlated with final histology, resulted in sensitivity ranging from 80–83.5% and specificity 71.4–74.3%, not outperforming cytology alone by much [148,149,150,151,152,153]. CD20 performed better as the diagnostic biomarker, especially in low-grade (sensitivity 64.3 and specificity 90.0%) [34,154,155], although some Authors found discordant results and suggested cautious interpretation of positive immunohistological staining [155]. On the other hand, it was argued not to use ki-67 when trying to distinguish CIS and florid atypia as it performed poorly [156]. 

The cytology can also be improved by performing **quantitative phase imaging** on diffraction phase microscopy by measuring nuclear/cell dry mass. In fact, patients in the suspicious group presented with significantly higher nuclear mass and entropy cells than controls [157]. Finally, staining for ARID1A using immunocytochemistry performed well as a predictive, but not as well as a diagnostic biomarker [158].

## 4. Conclusions

Patients with hematuria are a significant burden to urology clinics, and despite of the phenomenon frequency, no clear consensus exists on how to diagnose nor treat them. Usually, cystoscopy is performed. Furthermore, American and European guidelines recommend cystoscopy and cytology as a follow-up in non-muscle invasive BCa [3]. Concerns regarding poor sensitivity of cytology and frequent ‘inconclusive’ results [11,23,159,160], as well as not a negligible percentage of rare variants that are often even more difficult to be diagnosed and ambiguous flat lesions reported during cystoscopies, are more present than ever [21,22,23]. These were the main reasons why researchers have shifted their attention into discovering other, noninvasive, low cost, non-observer dependent, non-labor-intensive, rapid and accurate detection tests.

In this review we discussed a significant amount of potential urinary and serum diagnostic biomarkers for bladder cancer, the majority of them in need of further validation in larger, multicentric, blinded and prospective trials. Nevertheless, some have a great potential, with satisfactory negative predictive value, such as, e.g., assays containing multiple pathological gene expression [130]. The summary of all new potential biomarkers is shown in Table 1.

The problem we are facing now, as described in this review, is an enormous variability in sensitivity, specificity and NPV of the same biomarker. This, although justifiable by various factors such as age, sex, lifestyle, number of exfoliated cell in urine, high number of mutational burden and/or presence of hematuria, creates an important confusion between clinicians, those who actually need to use these tests. Moreover, a significant number of proposed biomarkers still needs an external validation, and this is the main reason why results reported in these studies may not be attainable. Another interesting fact to take in consideration is the significant heterogeneity of the bladder cancer, not only interpatient variability, but an intra-tumoral heterogeneity [159,160,161,162]. This means that, based on the exfoliated cells, even the most precise genetic tests could miss a diagnosis [159,161].

The most important factor a novel biomarker should have is a high NPV. The acceptable NPV should be higher than that of cystoscopy, therefore, at least >95% and it should not be operator dependent. In the past, FDA-approved tests (NMP22, BladderChek, UroVysion, EpiCheck, ImmunoCyt, BTA-TRAK and BTA-STAT) have been used, but recently, some decline in their clinical use has been observed, because there is a lack of clear indications on their use, additional cost, complicated processing or even availability problem.

It seems that the most appropriate test should be an assay, a collective of biomarkers that integrates the most frequently mutated genes, with common epigenetic changes, altered superficial proteins and tumor environmental factors, such as immunological response and urinary microbiome, as well as metabolic changes that are tumor induced. However, most importantly, NPV aside, it should be readily available even in the most peripheric urology clinic and it should be cost-efficient.

We reviewed the literature on urinary and serum diagnostic biomarkers present in the last 5 years, so some good biomarkers discovered and validated prior to 2016 may not be included in this paper, which poses a limitation. Nevertheless, we did try to simplify as much as possible the current biomarker landscape, not only to enrich the knowledge of the reader, but also to stimulate young clinicians to be more involved when it comes to novelty, especially if that means that money saved from unnecessary cystoscopies can be redirected to other, more useful channels.

**Table 1 ijerph-19-09648-t001:** Summary of all potential BCa diagnostic biomarkers.

Biomarker Category	Biomarker	Sensitivity	Specificity	References
	Conventional cytology	16–100%	86%	[3]
** Altered proteins **				
	Cytokeratins 8 and 18 (UBC rapid test)	30–87%	63–91%	[28,29,30,31,32]
	Cytokeratin 20	56–76%		[33,34]
	NMP22	37.9–88.5%	65.2–96.9%	[36]
	SPARC	39–43%	70–78%	[38]
	Orosomucoid-1	92%	94%	[45]
	APE1/REF1	81.7%	79.6%	[39]
	Soluble FAS	88.03%	89.19%	[40]
	AURKA	79.6%	79.7%	[41]
	Serum Irisin	74.7%	90.7%	[42]
	AIB1	80%	86%	[44]
	EIF5A2	74%	78%	[44]
	AIB1, EIF5A2, NPM22	89%	91%	[44]
	LASP1	59%	80%	[47]
	Unphosphorylated TF	70.6%	97.8%	[48]
	Phosphorylated TF-pSer258	88.2%	93.3%	[48]
	Phosphorylated TF-pSer253	88.2%	24.4%	[48]
	Multiplex immunoassay—A1AT, APOE, ANG, CA9, IL8, MMP9, MMP10, PAI1, SDC1, VEGFA	87–93%	93%	[49,50,51]
** Genomic mutations **				
	TERT promoter mutation	46.7–90%	90–100%	[56]
	TERT, FGFR3, KRAS	79–98%	62–90%	[57,58]
	TERT and PLEKHS1 promoters	91–95%	96–100%	[62]
	TERT, FGFR3, OTX1	72%	59%	[60]
	FEGFR3, Cyclin D3	73%	90%	[59]
	FGFR3, TP53, PIK3CA, ARID1A, STAG2, KTM2D	73–95%	85–90%	[61]
	ASPM upregulation	95.56%	86.67%	[63]
	TEF downregulation	91.11%	71.11%	[64]
	MCM5 overexpression	63–79%	65–88%	[65,66,67,68,69,70]
	Microsatellite instability (MIS) or Loss of heterozigosity (LOH) of D16S476, D9S171, FGA, ACTBP2	96.7%	30%	[72]
** Circulating free nucleid acids **				
	Urine-derived fc-DNA	20.7%	91.2%	[74]
	Urine exosomes	92.6%	83.3%	[75]
	Serum exosomes	82.4%	100%	[75]
	5 mRNA mutations panel: MDK, HOXA13, CDC2, IGFB5, CXCR2 (Cxbladder)	91–97.7%	61–85%	[76,77,78,79]
	5 mRNAs: CRH, IGF2, UPK1B, ANXA10, ABL1 (Xpert Bladder)	63–80%	73–81%	[80,81,82,83,84,85,86,87,88,89]
	UBEC2	82.5%	76.2%	[90]
	IQGAP3	80–96.2%	60.2–90.7%	[91]
	N-Myc	85.5%	81.4%	[92]
** Epigenetic changes and Transcriptomic alterations **				
	miRNA 130 family (-130a-3p, -130b-3p, -301a-3p)	87.8%	93.3%	[102]
	miRNA 192	76.7%	78%	[100]
	miRNA 192, 2D ultrasound	93.2%	96.7%	
	2 miRNAs panel: -31-5p, -93-5p	82%	70%	[99]
	7 miRNAs panel: -7-5p, -22-3p, -29a-3p, -126-5p, -200a-3p, -375, -423-5p	80.88%	91.67%	[98]
	12 miRNAs panel: -16, -21, -34a, -99a, -106b, -126, -129, -133a, -145, -200c, -205, -218, -221/222, -331	88%	48%	[103]
	miRNA urinary supernatant: -125b, -30b, -204, -99a, -532-3p	59%	96%	[97]
	7 serum miRNAs panel: -6087, -6724, -3960, -1343-5p, -1185-1-3p, -6831-5p, -4695-5p	95–98%	87–91%	[101,104]
	25 miRNAs panel (-140-5p, -142-5p, -199a-3p, -93, -652, -20a, -106b, -1305, -223, -18a, -191, -126, -26b, -26a, -145, -146a, -30a-3p, -96, -573, -221, -182,-142-3p, -19b, -224, -181a, -766, -146b-5p, -429, -200a, -200c, -20b, -324-3p, -19a, -106a, -143, -99b, -140-3p, -491-5p, -151-3p, -671-3, -222, -339-3p, -141, -200b, -7b, -21	87%	100%	[95]
	10 miRNAs panel: -652, -199a-3p, -140-5p, -93, -142-5p, -1305, -30a, -224, -96, -766	84%	87%	
	DNA methylation test of 15 (unpublished) genes (Bladder EpiCheck)	57–85%	80–88%	[106,107,109,110,111]
	Methylation of tumor suppressor genes (p14ARF, p16INK4A, DAPK, RASSF1A, APC)	91–100%	NA	[117]
	DNA hypermethylation of 150 loci panel (UroMark)	96%	97%	[118,119]
	Hypermethylation of OTX1, ONECUT2, TWIST1, SEPTIN9, PCDH17, POU4F2, HS3ST2, SLIT2, FGFR3, CFTR, SALL3, GHSR, MAL, mutation of HRAS, TERT, FGFR3	93–98%	40–86%	[60,112,113,114,115,116]
	RNA sequencing of two cancer specific-genes and one reference gen: ROBO1, WNT5A, CDC42BPB	83%	89%	[121]
	Urinary lncRNA uc004cox.4	80%	85%	[125]
	3 lncRNAs panel: PCAT-1, UBC1 and SNHG16	80%	75%	[122]
	3 lncRNAs panel: MALAT1, PCAT1, SPRY4-IT1	62.5%	85%	[123]
	3 lncRNAs panel: MALAT1, MEG, SNHG16	82%	73%	[124]
	3 lnc RNAs panel: PVT-1, ANRIL, PCAT-1 (tested only in T1-T2)	46.67%	87.5%	[126]
	7 lncRNAs panel: HOTAIR, NEAT1, TUG1, FAS-AS1, PVT1, GHET1, HOTAIRM1	100%	100%	[127]
	4 lncRNAs panel: LINC00355, UCA1-201, UCA1-203, MALAT1	92%	91.7%	[128]
	3 lncRNAs panel: H19, UCA1 and HOTAIR	70.8%	88.5%	[129]
	12 genes panel transcriptomic alterations: IGF2, MAGEA3, KLF9, CRH, SLC1A6, POSTN, TERT, AHNAK2, ANXA10, CTSE, KRT20, PPP1R14D	79%	93%	[130]
	10 genes panel transcriptomic alterations: IGF2, MAGEA3, KLF9, CRH, SLC1A6, POSTN, EBF1, CFH, MCM10, MMP12	80%	94%	
	5 genes panel transcriptomic alterations: IGF2, MAGEA3, KLF9, CRH, SLC1A6	79%	92%	
	2 genes transcriptomic alterations: IGF2, MAGEA3	81%	91%	
** Metabolomics and Metabonomics **				
	Dimethyl amine, malonate, glutamine, lactate, histidine and valine metabolites panel	80.8–98.1%	66.7–80.3%	[133]
	Phosphatidylinositol, nucleic acids, collagen, aromatic amino acids, cholesterol fatty acids, glycogen, monosaccarides and carotenoids’ changes (Rametrix)	82.4%	79.5%	[134]
	Bladder wash (>resectisol, urea, creatinine, uric acid, different types of cells, cylinders, crystals) analyzed by FTIR	81.8–100%	52.9–80.9%	[135]
	6 ions panel (>imidazoleacetic acid)	82%	85–90%	[136]
	116 peptides panel (>collagen fragments, APO-I peptides, basement-membrane specific heparan proteoglycan fragments)	88–91%	51–68%	[138]
	3 serum metabolites panel: Inosine, PS(O-18:0/0:0), Acetyl-N-formyl-5-methoxykynurenamine	84.6%	84.6%	[137]
	Concentration matrices	55%	74.7%	[139]
** Combination of more than one category **				
	miRNAs: -19b1-5p, 21-5p, 136-3p, -139-5p, 210-3p combined with BLCA-4, NMP22, APE1/Ref1, CRK, VIM, and creatinine urinary concentrations	80%	88%	[46]
	miRNA-663, VIM hypermethylation	92.6%	90%	[141]
	Altered methylation of CFTR, SALL3, TWIST 1, NID2, TWIST1 in adjunct to Cytology	57–96%	40–72%	[142,143]
	Altered methylation of SALL3,ONECUT2, CCNA1,BCL2, EOMES, VIMcombined with altered mutations of TERT, FGFR3	97%	77%	[144]
	3 mRNAs: KLHDC7B, CASP14 and PRSS1; lncRNAs: MIR205HG and GAS5	87.2%	83.3%	[145]
	Metabolomics (CRAT and SLC 25A20 genes expression) + 5 mRNAs: CRH, IGF2, UPK1B, ANXA10, ABL1	66%	98%	[146]
	MMP-2, MMP-9 and TIMP-2 urinary and serum proteins’ and genes’ expression levels	100%	100%	[147]
** Improving cytology **				
	Cytology and CK20 immunostaining	64.3–97%	80–90%	[150,154]
	Cytology with CK20 and p53 immunostaining	91.1%	74.3%	[149,156]
	Cytology enriched with p53, ki67 coloration	68.9%	97.5%	[148,155]
	Cytology combined with AMARC coloration	73%	97%	[150]
	Cytology combined with p16/ki-67 dual-labeling	80%	71.4%	[151]
	p53, MCM5, MCM2, ki-67 coloration in adjunct to cytology	67.3–90.4%	72–80%	[153]
	Cytology combined with loss of ARID1A expression	NA	NA	[158]

## Figures and Tables

**Figure 1 ijerph-19-09648-f001:**
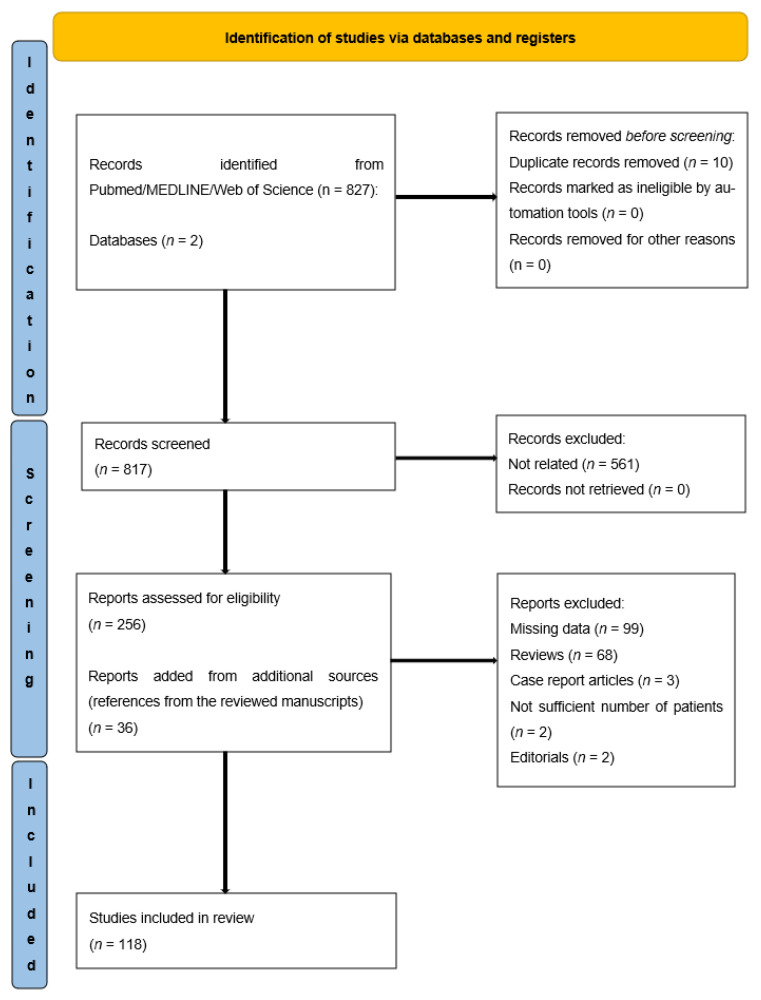
PRISMA flowchart.

## Data Availability

Not applicable.

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
