# Peer review of "The Role of Novel Bladder Cancer Diagnostic and Surveillance Biomarkers—What Should a Urologist Really Know?"

_ijerph, 2022, doi:10.3390/ijerph19159648_

Round 1

Reviewer 1 Report

The review is well written and highlights some of the major biomarker discoveries and developments in the field of bladder cancer. The review is written such that any clinician can understand the current research and developmental efforts in the area. The manuscript has a high impact not only on researchers but also aids as a tool for clinicians. The reviewer would like to suggest a few minor comments to be addressed.  

1. Figure 1 is cut off at certain locations which need to be adjusted and requires a legend. Additionally, in the legend please include what ** provided in the figure for records excluded denotes.

2. Provide the full abbreviations when appearing for the first time in the text.

3. Even though, the text talks about the sensitivity and specificity of cytology only; it would be beneficial for readers if this info is provided in the table for easier comparison to the cited work.

Author Response

Reviewer's comments:

The review is well written and highlights some of the major biomarker discoveries and developments in the field of bladder cancer. The review is written such that any clinician can understand the current research and developmental efforts in the area. The manuscript has a high impact not only on researchers but also aids as a tool for clinicians. The reviewer would like to suggest a few minor comments to be addressed.  

  1. Figure 1 is cut off at certain locations which need to be adjusted and requires a legend. Additionally, in the legend please include what ** provided in the figure for records excluded denotes.
  2. Provide the full abbreviations when appearing for the first time in the text.
  3. Even though, the text talks about the sensitivity and specificity of cytology only; it would be beneficial for readers if this info is provided in the table for easier comparison to the cited work.

Dear Reviewer and Editor,

Firstly, we would like to thank you for your time dedicated to our manuscript, and for the suggestions made in order to improve it. Also, we appreciated greatly your kind comments.

  1. We adjusted Table 1. following your suggestions.

  1. As advised, everyfirst time we used an abbreviation, we spelt out the full term and put the abbreviation in parentheses.

  1. We agreed the proposition you made and added the sensitivity and specificity of the urine cytology to the Table 1.

We truly hope that the revised manuscript is now clarified enough.

Again, we deeply appreciate your help on improving the readability of our paper.

The Authors

Reviewer 2 Report

In the manuscript “The role of novel bladder cancer diagnostic and surveillance biomarkers – what should a urologist really know?”, Malinaric et al. analyzed recently published biomarker-related studies of bladder cancer and identified a list of potential biomarkers. This study updated our knowledge on bladder cancer biomarkers, which is interesting and clinically relevant. However, there are minor changes needed before the manuscript meets publication quality.

1.     The authors need to improve writing of the entire manuscript. There are some grammar error and part of the manuscript is hard to follow. For example, line 16, divided them in <- divided them into.

2.     The format of Figure 1 needs to be improved and the figure legend is missing.

3.     For Table 1, how Sensitivity and Specificity values are calculated need to be described in the method section.

Author Response

Reviewer's comments:

In the manuscript “The role of novel bladder cancer diagnostic and surveillance biomarkers – what should a urologist really know?”, Malinaric et al. analyzed recently published biomarker-related studies of bladder cancer and identified a list of potential biomarkers. This study updated our knowledge on bladder cancer biomarkers, which is interesting and clinically relevant. However, there are minor changes needed before the manuscript meets publication quality.

  1. The authors need to improve writing of the entire manuscript. There are some grammar error and part of the manuscript is hard to follow. For example, line 16, divided them in <- divided them into.
  2. The format of Figure 1 needs to be improved and the figure legend is missing.
  3. For Table 1, how Sensitivity and Specificity values are calculated need to be described in the method section.

Dear Reviewer and Editors,

Firstly, we would like to thank you for your time dedicated to our manuscript, and for the suggestions made in order to improve it. Also, we appreciated greatly your kind comments.

  1. The entire manuscript was edited for proper English language. We tried to better the structure of the sentences and paragraphs as well.
  2. We adjusted the Figure 1. as you suggested.
  3. All reported sensitivities and specifities were extracted from the results of the reviewed articles. We did, however, add that information in the results section (please, see the line 96.).

We truly hope that the revised manuscript is now clarified enough.

Again, we deeply appreciate your help on improving the readability of our paper.

The Authors

Reviewer 3 Report

In this study, Malinaric et al. performed an extensive review of publications on bladder cancer biomarkers from 2016 to 2022, categorized the biomarkers into different application groups, and discussed their predictive values. This is a good adding to current knowledge in the bladder cancer field. It is also interesting to readers who pay attention to biomarker development.

While this paper is well written, there are a few unclear or confusing parts in the manuscript, which need thorough revisions before further consideration.

1. The authors classified biomarkers into 6 categories: Altered proteins, Genomic mutations, Epigenetic changes and transcriptomic alterations, Metabolic-induced changes, Combination of more than one category and Improving conventional cytology. This is smart but the contents later in the RESULTS section did not fully agree with the categories.

For example, Microsatellite in-stability is usually considered as a marker for genomic mutation. The cell-free DNA or RNA is not recommended to put into genomic mutation category. Instead, it can be set as an independent category. The mRNA examples under genomic mutations are confusing because there seem to be both mRNA mutation biomarker and mRNA level biomarker, which overlaps with latter transcriptomic alterations.

The authors should re-organize and finetune their sub-categories.

2. The authors stated that 36 articles were added from additional sources but did not specify the information of the sources. And the numbers of articles included did not fully agree with the numbers indicated in Figure 1. Also, Figure 1 has multiple missing letters and words in the PDF version. The authors should have checked the format before submission.   

3. There are a few unexplained abbreviations such as CIS, NPV, PRISMA. Although the authors may assume the terms well known by the target readers, it would be better if they put the full name at the first appearances of the abbreviations.

Author Response

Reviewer's comments:

In this study, Malinaric et al. performed an extensive review of publications on bladder cancer biomarkers from 2016 to 2022, categorized the biomarkers into different application groups, and discussed their predictive values. This is a good adding to current knowledge in the bladder cancer field. It is also interesting to readers who pay attention to biomarker development.

While this paper is well written, there are a few unclear or confusing parts in the manuscript, which need thorough revisions before further consideration. 

  1. The authors classified biomarkers into 6 categories: Altered proteins, Genomic mutations, Epigenetic changes and transcriptomic alterations, Metabolic-induced changes, Combination of more than one category and Improving conventional cytology. This is smart but the contents later in the RESULTS section did not fully agree with the categories.

For example, Microsatellite in-stability is usually considered as a marker for genomic mutation. The cell-free DNA or RNA is not recommended to put into genomic mutation category. Instead, it can be set as an independent category. The mRNA examples under genomic mutations are confusing because there seem to be both mRNA mutation biomarker and mRNA level biomarker, which overlaps with latter transcriptomic alterations.

The authors should re-organize and finetune their sub-categories.

  1. The authors stated that 36 articles were added from additional sources but did not specify the information of the sources. And the numbers of articles included did not fully agree with the numbers indicated in Figure 1. Also, Figure 1 has multiple missing letters and words in the PDF version. The authors should have checked the format before submission.   
  2. There are a few unexplained abbreviations such as CIS, NPV, PRISMA. Although the authors may assume the terms well known by the target readers, it would be better if they put the full name at the first appearances of the abbreviations.

Dear Reviewer and Editor,

Thank you very much for the time you dedicated to our manuscript, and for all the insightful comments. We read carefully your suggestions, and did our best to better the article.

  1. Firstly, we reogranized some paragraphs and added the whole new category as follows:

As you correctly noticed, microsatellite instability is a genomic biomarker, not an epigenetic nor transcriptomic. For that reason we moved the whole paragraph in the appropriate category.

As suggested, we created the whole new category dedicated only to free-cell DNA and RNA biomarkers. Subsequently, we kept the mRNA level biomarkers in the 'Circulating free nucleic acids' category, whilst moved the paragraph dedicated to the mRNA mutational biomarkers at the beginning of 'transcriptomic alterations' category. We apologize for all the above mentioned inaccuracies.

  1. We are sorry for the format of the PRISMA flowchart in the previous version of the manuscript. We specified where additional resources came from in the 'Results' section (please, see the line 109. and PRISMA flowchart). Furthermore, we reorganized the flowchart and made it, hopefully, more comprehensible.

  1. As advised, every first time we used an abbreviation, we spelt out the full term and put the abbreviation in parentheses.

We truly hope that the revised manuscript is now clarified enough.

Again, we deeply appreciate your help on improving the readability of our paper.

The Authors.
